# Self-Diagnostic and Self-Compensation Methods for Resistive Displacement Sensors Tailored for In-Field Implementation [note 1]

**DOI:** 10.3390/s24082594

**Published:** 2024-04-18

**Authors:** Federico Mazzoli, Davide Alghisi, Vittorio Ferrari

**Affiliations:** 1Department of Information Engineering, University of Brescia, Via Branze 38, 25123 Brescia, Italy; vittorio.ferrari@unibs.it; 2Gefran SpA, Via Cave 11, 25050 Provaglio d’Iseo, Italy; davide.alghisi@gefran.com

**Keywords:** smart sensor, self-validating sensor, resistive displacement sensors

## Abstract

This paper presents a suitably general model for resistive displacement sensors where the model parameters depend on the current sensor conditions, thereby capturing wearout and failure, and proposes a novel fault detection method that can be seamlessly applied during sensor operation, providing self-diagnostic capabilities. On the basis of the estimation of model parameters, an innovative self-compensation method is derived to increase the accuracy of sensors subject to progressive wearout. The proposed model and methods have been validated by both numerical simulations and experimental tests on two real resistive displacement sensors, placed in undamaged and faulty conditions, respectively. The fault detection method has shown an accuracy of 97.2%. The position estimation error is < ±0.2% of the full-scale span for the undamaged sensor, while the self-compensation method successfully reduces the position estimation error from ±15% to approximately ±2% of the full-scale span for the faulty sensor.

## 1. Introduction

Displacement sensors are widely used in different applications and fields, such as industrial [1,2], medical [3,4] and automotive [5,6]. Within the industrial field, the role of displacement sensors in control applications is gaining importance since the degree of industrial automation is continuously growing. In this context, there is a constant interest in displacement sensors with increasing intelligence, robust measurement methods, and improved performances [7]. In particular, sensor reliability in producing accurate measurement data is crucial. Augmenting sensor diagnostic capabilities plays an increasingly important role, as undetected failures may negatively affect the industrial process into which the sensor is inserted, resulting in efficiency loss, downtimes or even threatening life safety [8].

Several solutions can be considered to prevent sensor failure and enhance sensor diagnostic coverage. A first approach is the adoption of a Preventive Maintenance (PM) policy, which implies sensor calibration and maintenance activities run on a regular time basis to avoid failures before they occur [9]. Determining in advance when a sensor will enter the wearout phase is challenging, as predictions typically rely on a theoretical failure rate rather than on the current sensor conditions. While PM can reduce unexpected process downtime, it can lead to costly and unnecessary procedures and repairs.

A second possible approach involves employing Fault Detection and Identification (FDI) theory [10,11,12,13]. FDI techniques recognize system faults in process components by monitoring system inputs and outputs in order to generate residuals highlighting discrepancies. Thresholds, determined through experimental tests, aid in fault symptom identification. Analytical redundancy techniques [14,15,16] detect and identify sensor failures from the exploitation of analytical models built on a specific process, which includes actuators and plants as well as sensors. However, FDI techniques require a deep knowledge of the entire process and a significant effort in model development. Furthermore, the model must be updated as the process changes. The economic effort conveyed in the model design is not easily amortized since the model is related to a specific process. Moreover, technical staff must be available for maintaining and updating the model as the process evolves over time.

A third possible approach to detect and identify sensor faults is through the use of Machine Learning (ML) techniques [17,18,19]. The ML models are trained on sensor data to identify patterns or features indicative of faults. The performance of ML techniques are dependent on the quality and quantity of the available training data. In fact, the training data could be limited, biased, or not representative of all possible faults, which can limit the detectable faults. Moreover, the ML approach mainly focuses on detecting the sensor faults through certain time and/or frequency domain signal characteristic features. In particular, faults can be identified by changes in signal behavior, e.g., bias, noise, spikes, rather than by identifying underlying sensor-specific causes for the fault [20].

An alternative approach could be the adoption of smart sensors inherently capable of providing self-diagnostic information related to the alteration of their conditions due to incipient wearout or failure. The diagnostic information provided might be exploited by suitable Predictive Maintenance (PdM) techniques to optimize sensor maintenance scheduling and minimize plant downtime [21]. As opposed to PM approaches, self-diagnostic sensors could avoid both unnecessary maintenance costs and plant downtime due to undetected sensor failures. Moreover, in contrast to many FDI techniques, self-diagnostic sensors place the focus within the sensor itself to make it a more reliable component, leaving out the additional system parts of a given industrial process. Therefore, in principle, a generic process can be freely modified without requiring any change in the diagnostic techniques embedded in the self-diagnostic sensor. As such, the research on displacement sensors equipped with self-diagnostic functionalities is a topic of relevant importance, which, however, has been explored to a somewhat limited extent. In particular, resistive displacement sensors would benefit from solutions for embedding self-diagnostic capabilities. However, to the authors’ best knowledge, the current state of the art does not yet adequately cover the development of a resistive displacement sensor model for self-diagnostic and self-compensation purposes.

In this context, this work innovatively proposes a simple yet effective model for a generic resistive linear displacement sensor where the model parameters directly depend on the current sensor conditions, thereby capturing wearout, damaging, and failure occurrence. Furthermore, a method for the continuous estimation of the model parameters is presented. The method originally offers the advantage that it can be applied while the sensor is kept in operation online, i.e., without functionally disconnecting the sensor from the monitored process. The estimated parameters are exploited to detect failures by the proposed fault detection technique, which aims at detecting faults by identifying the underlying sensor-specific cause. The continuously updated parameters are used by the self-compensation method to compensate for the position estimation error arising over time due to progressive wearout and cumulative damage. This paper considerably extends the preliminary work previously presented [22], deepening the study and analysis of both the model for a resistive linear displacement sensor and related self-diagnostic and self-compensation methods.

A schematic comparison between the most common methods to detect and identify sensor faults and the approach proposed in this work is reported in Table 1.

The paper is organized as follows. A generic resistive linear displacement sensor model along with the proposed parameter estimation method, method for compensating position estimation errors, and fault detection method are described in Section 2. In Section 3 the numerical simulations carried out and the experimental setup arranged to test the proposed model and methods are described and the results are reported. The obtained experimental results are discussed in Section 4. The conclusions are drawn in Section 5.

## 2. Materials and Methods

### 2.1. Resistive Displacement Sensor Model

Resistive displacement sensors measure the linear or angular displacement of an object by sensing a resistance change related to the object position variation. With specific reference to linear displacement sensors along the direction *x*, as shown in Figure 1a, the sensing element is essentially composed of a resistive track, a sliding cursor, which is constrained to move along the longitudinal axis of the sensor, and a conductive track. The resistive and conductive tracks are electrically connected to each other through the sliding cursor, which is mechanically linked to the moving object under measurement.

The most common causes of failure for linear sensors include the following:Wear: sensor components like the sliding cursor and the resistive/conductive track can wear out or become contaminated with dust, dirt, or other debris, causing the sensor to malfunction or eventually fail.Mechanical damage: shock, vibrations or impacts can damage or even break the sliding cursor or the resistive/conductive track.Environmental factors: the sensor can be affected by temperature changes or gradients, pressure variations impairing sensor sealing, moisture infiltration, or other causes producing changes in resistance, which leads to measurement errors up to complete failure.Aging: the electrical properties of materials forming the resistive/conductive track and the cursor might change over time, causing sensor resistance drift or output instability, leading to inaccurate measurements up to an unacceptable level.

Considering the above-mentioned failure modes, a tailored sensor model is presented below to provide information about the actual sensor conditions and its possible alterations over time. As shown in Figure 1b, the sensing element is modeled by a resistive circuit composed of three resistors. The resistor *R*_c_ represents the contact resistance between the sliding cursor and the resistive track. The resistors *R*_p1_ and *R*_p2_ represent the resistive track resistances originated from its partitioning caused by the sliding cursor according to its position. Resistances *R*_p1_ and *R*_p2_ as a function of the sliding cursor position *x* are given by the following:*R*_p1_ = (*ρ/S*)*x* = ξ*x*(1)
*R*_p2_ = (*ρ/S*)(*L* − *x*) = ξ(*L* − *x*)(2)
where *ρ*, *S*, *L*, and ξ = *ρ/S* are the electrical resistivity, section area, length, and resistance per unit length of the resistive track, respectively. The conductive track is assumed to have zero resistance.

The resistances measured between the sensor terminal pairs (1, 2), (2, 3), and (1, 3) can be expressed as follows:*R*_12_ = *R*_p1_ + *R*_c_ = ξ*x* + *R*_c_(3)
*R*_23_ = *R*_c_ + *R*_p2_ = *R*_c_ + ξ(*L* − *x*)(4)
*R*_13_ = *R*_p1_ + *R*_p2_ = ξ*x* + ξ(*L* − *x*) = ξ*L*(5)

Equations (3) and (4) can be combined obtaining the following:Σ = *R*_12_ + *R*_23_ = ξ*x* +*R*_c_ + *R*_c_ + ξ(*L* − *x*) = 2*R*_c_ + ξ*L*(6)
where the term Σ is independent from the sliding cursor position *x* at which the resistances *R*_12_ and *R*_23_ are measured.

Therefore, the sensor model is expressed by combining Equations (5) and (6) in matrix form as follows:(7a)$$Ap=R\to \left[\begin{array}{cc} 2 & L\\ 0 & L \end{array}\right]\left[\begin{array}{c} R_{c}\\ \xi \end{array}\right]=\left[\begin{array}{c} \Sigma \\ R_{13} \end{array}\right],$$
(7b)$$p=A^{-1}R\to \left[\begin{array}{c} R_{c}\\ \xi \end{array}\right]=\left[\begin{array}{c} \frac{1}{2}\Sigma -\frac{1}{2}R_{13}\\ {\frac{1}{L}R}_{13} \end{array}\right]=\left[\begin{array}{c} \frac{1}{2}\left(R_{12}+R_{23}\right)-\frac{1}{2}R_{13}\\ {\frac{1}{L}R}_{13} \end{array}\right]$$
where the vector ***R*** is formed by the resistance terms Σ and *R*_13_, the vector ***p*** includes the sensor parameters *R*_c_ and ξ, and the square matrix ***A*** only depends on the resistive track length *L*. Notably, the linear model of Equation (7a), expressed in its inverted form in Equation (7b), is independent from the measurand displacement *x*. As such, it advantageously allows to estimate ***p*** by only knowing the resistive track length *L* and the resistances *R*_12_, *R*_23_, and *R*_13_ measured at any unknown cursor position *x*.

The parameters *R*_c_ and ξ offer the advantage to directly relate the measurable sensor electrical characteristics with the elements that are most susceptible to wear and faults, i.e., the sliding cursor and resistive track [23]. Although more parameters could be taken into consideration to obtain a more detailed sensor model, *R*_c_ and ξ already allow the identification of the most common causes of wearout and failure for resistive displacement sensors, at the same time avoiding excessive model complexity. In addition, the estimation of *R*_c_ and ξ can be advantageously performed without the knowledge of the cursor position *x*, allowing their continuous estimation during sensor normal operation.

The parameters ***p*** are used to detect alterations in sensor conditions up to sensor faults. For example, the wearout caused by the contact force between the sliding cursor and the resistive track can be considered. Repeated cycles of slides and/or an excessive contact force leads to a reduction in resistive track thickness by abrasion, and in turn a reduction in section *S*, with possible overheating and local oxidation and wear [24]. Compared to the initial conditions of a brand-new sensor, the reduction in *S* implies an increase in ξ. The resistive element wear debris deposited between the sliding cursor and the resistive element increases the contact resistance *R*_c_ because the contact area decreases, and the debris acts as an insulator. In addition, creep and fatigue in the sliding cursor brush cause the contact force to decrease, again increasing *R*_c_ [23,25]. With respect to possible mechanical damage, the breakout of the sliding cursor opens the electrical contact between the resistive and conductive tracks, leading *R*_12_ and *R*_23_ to infinity, as well as *R*_c_. Other possible mechanical damages could be cracks in the resistive track, which typically increase the resistance *R*_13_ and thus ξ. Additionally, the resistive track ages, leading to a change in resistivity *ρ*, which reflects in a change of ξ. Meanwhile, the slider cursor ages by oxidizing, causing *R*_c_ to increase.

The model assumes operation at constant temperature. Such an assumption is not a substantial limitation in practice, as in the majority of industrial applications, the sensor, after a thermal transient at the machine start-up, works at a reasonably constant temperature thereof. For this reason, though track resistivity may in general depend on temperature, the proposed model can be conveniently restricted to thermal steady-state operation without suffering unacceptable inaccuracies.

Sensor condition alterations and failures are detected by monitoring the evolution of *R*_c_ and ξ with respect to their initial values, i.e., those of a brand-new undamaged sensor. Initial values of *R*_c_ and ξ are design parameters subject to the typical variability of the manufacturing process. In the first analysis, the detection of fault conditions can be accomplished by comparing each parameter with a respective nominal value taken as the alarm threshold. Due to product and process variability, the definition of *a priori* nominal alarm threshold could be ineffective for diagnostic monitoring of sensor conditions. For this reason, it is proposed to identify specific alarm thresholds for each individual sensor.

### 2.2. Parameter Estimation

To determine the initial parameters and monitor their evolution during the sensor operation, the proposed estimation method derives the parameters of the vector ***p*** online, i.e., without functionally disconnecting the sensor from the monitored process, and without knowing the cursor position *x*.

Online estimation is a substantial advantage of the proposed method and is mandatory for its effective practical application since both the initial parameters and their variations during operation in the field must be determined.

As illustrated in the flow chart of Figure 2, the estimation method can be divided into two phases. In the first phase, denoted as the measuring phase, the resistance set represented by the three-component vector ***S*_R_** = [*R*_12_, *R*_23_, *R*_13_] measured at an unknown cursor position *x* is used to evaluate the resistance vector ***R*** = [Σ, *R*_13_]. In the second phase, denoted as the estimation phase, the previously evaluated ***R*** is used to compute ***p*** as reported in Equation (7b).

In the proposed method, each resistance set ***S*_R_**, measured at an unknown position *x*, is used to derive a correspondent estimation of ***p***. The estimated parameters reflect the sensor local conditions at the unknown position *x* where ***S*_R_** is measured. This provides indirect indications of possible nonhomogeneities.

As an alternative, ***p*** can be computed using the average of *N* resistance sets ***S*_R_** measured in *N* different cursor positions. Provided that the *N* positions are sufficiently spread, this results in spatial averaging along the resistive track with a correspondent reduction the residual dependence of p on the cursor position *x*.

In this work, the parameters p are estimated starting from each single resistance set ***S*_R_**, i.e., *N* = 1. This offers the advantage of maintaining a biunivocal correspondence between ***p*** and the position *x* without introducing spatial averaging.

Since the parameters ***p*** are estimated starting from the direct measurement of the resistances *R*_12_, *R*_23_, and *R*_13_, the lower the measurement uncertainty of the resistance measurement, the better the parameter estimation will be. The uncertainties *u*_c_(*R*_c_) and *u*_c_(ξ) resulting from the resistance uncertainties *u*(*R*_12_), *u*(*R*_12_), and *u*(*R*_13_) can be considered in defining a fault detection method that is robust with respect to measured data. From Equation (7b), the composite uncertainties *u*_c_(*R*_c_) and *u*_c_(ξ) are given by the following:(8)$$u_{c}\left(R_{c}\right)=\sqrt{\frac{1}{4}u^{2}\left(R_{12}\right)+\frac{1}{4}u^{2}\left(R_{23}\right)+\frac{1}{4}u^{2}\left(R_{13}\right)}$$
(9)$$u_{c}\left(\xi \right)=\sqrt{\frac{1}{L^{2}}u^{2}\left(R_{13}\right)}$$

### 2.3. Cursor Position Estimation and Compensation

There are several methods to obtain the cursor position of a resistive displacement sensor. Considering the schematization of a generic resistive displacement sensor of Figure 1a, one common method involves exciting the sensor with a known constant voltage at the terminal pair (1, 3) while reading at high-input impedance the voltage divider output at the terminal pair (2, 3), which varies as a function of the cursor position *x*. This method does not account for the sensor conditions represented by parameters ***p***. Actually, the reading is to first order independent from ***p*** as long as ξ is uniform along the sensor stroke *L*. As no information on the sensor conditions is extracted, sensor faults cannot be anticipated nor predicted.

On the other hand, the resistances *R*_12_ and *R*_23_ depend on both the cursor position *x* at which they are measured and the parameters ***p***. By manipulating Equations (3) and (4), the actual cursor position *x* can be expressed as follows:*x* = (*R*_12_ − *R*_23_)/2ξ + *L*/2(10)

An estimated cursor position *x’* can be derived from Equation (10) by employing the current parameter estimates ***p*_c_** = [*R*_cc_, ξ_c_]. The closer the match between ***p*_c_** and ***p***, the lower the difference will be between *x’* and *x*. The position estimation error *e*_pos_ due to the difference between the actual unknown ξ and its current estimation ξ_c_ is as follows:(11)$$e_{pos}=x\prime -x=\frac{R_{12}-R_{23}}{2}\left(\frac{\xi -\xi_{c}}{\xi_{c}\xi }\right)$$
where *R*_12_ and *R*_23_ are the resistances measured in the current sensor conditions. Since the parameters vary as the sensor conditions change, it is crucial that the current parameters ***p***_c_ are continuously updated since *x’* depends on ξ_c_.

If instead of ***p*_c_** the nominal parameters ***p*_0_** = [*R*_c0_, ξ_0_] relative to an undamaged sensor are used as estimates of ***p*** during the sensor life, the estimated position $x_{0}\prime$ is subject to an estimation error *e*_pos0_ given by the following:(12)$$e_{pos0}=x_{0}^{\prime }-x=\left(x_{0}^{\prime }-x^{\prime }\right)+\left(x^{\prime }-x\right)=\frac{R_{12}-R_{23}}{2}\left(\frac{\xi_{c}-\xi_{0}}{\xi_{c}\xi_{0}}\right)+e_{pos}=\delta +e_{pos}$$

The position estimation error *e*_pos0_ can be seen as the sum of *e*_pos_ given by Equation (11) and an additive error δ related to the difference between the current parameter ξ_c_ and ξ_0_ employed for estimating position.

The additive error *δ* is zero when ξ_c_ = ξ_0_, i.e., when ξ does not vary from the undamaged conditions. It also shows that the error is higher the farther the cursor is from *L*/2. As the position error *e*_pos_ does not depend on *R*_c_ as long as *R*_c_ is finite and Equation (11) remains valid, the proposed estimation method is unaffected by *R*_c_.

Starting from Equation (12), if the estimate of ξ is constantly updated, the position estimation error *e*_pos0_ can be reduced to *e*_pos_ by nulling the additive position estimation error δ. In this perspective, the method proposed to reduce the position estimation error can be seen as a self-compensation method.

### 2.4. Fault Detection

The parameters *R*_c_ and ξ are known to play a crucial role in defining the conditions of the resistive displacement sensor as they are directly affected by sensor wearout and faults [23]. Therefore, the proposed fault detection method focuses on assessing progressive sensor wearout or sudden failure by comparing the variation in *R*_c_ and ξ with reference values by means of alarm thresholds. Innovatively, this approach is built on a model that allows determination of the sensor conditions during its operation, without disconnecting it from the host system and without requiring knowledge of the cursor position *x*.

As illustrated in the flow chart of Figure 3, faults are detected on the basis of parameter variations **Δ*p*** = [Δ*R*_c_, Δξ] = |***p*_c_** − ***p*_0_**|, where ***p*_c_** = [*R*_cc_, ξ_c_] and ***p*_0_** = [*R*_c0_, ξ_0_] represent the sensor parameters in the current and initial undamaged conditions, respectively. The nominal parameters ***p*_0_** are calculated as the spatial average of the initial parameters *R*_c_ and ξ estimated along the entire sensor stroke. The parameters ***p*_0_** could be determined during factory calibration or directly in the field, during the first installation. It is expected that *R*_c_ and ξ variations are mostly related to sliding cursor and resistive track faults, respectively.

Fault detection is based on threshold surpassing. Specifically, if both Δ*R*_c_ and Δξ are lower than their respective alarm thresholds *T*_1_ and *T*_2_, then no fault is detected. Differently, the sensor is faulty, and the fault could affect the resistive track, the sliding cursor, or both depending on which threshold is surpassed.

The thresholds *T*_1_ and *T*_2_ can be set empirically on the basis of ensemble standard deviations σ(*R*_c0_) and σ(ξ_0_) derived from a suitably large population representative of undamaged sensors, or from each individual sensor at the installation time in the field. Assuming that model parameters for an undamaged sensor are random variables with normal distribution, it is reasonable to suppose that changes that exceed the range of ±3σ(*R*_c0_) and ±3σ(ξ_0_) are ascribable to an occurred alteration in the sensor conditions due to wearout or fault.

## 3. Results

The sensor model, along with the parameter estimation, the self-compensation, and the fault detection methods have been simulated and experimentally tested.

The simulations have estimated the parameters from a set of resistance values corrupted with pseudorandom numerical noise, while in the experimental tests the parameters have been derived from measurements on a Device Under Test (DUT), i.e., a resistive displacement sensor, placed in two different conditions, namely undamaged and faulty.

The term undamaged here refers to a sensor that has never been used and has been stored in compliance with the requirements described in the sensor datasheet. Instead, the term faulty here indicates a sensor subject to a resistive track and sliding cursor wearout, where the resistive track wear along *x* is virtually uniform. Wear has been considered among the failures listed in Section 2.1 as it represents the main and most common failure mode.

The simulations have been aimed at evaluating the parameter estimation variability with respect to the measurement accuracy of resistances *R*_12_, *R*_23_, and *R*_13_. The experimental tests have been aimed at assessing the fitting accuracy of the sensor model defined as the difference between the measured resistances *R*_12_, *R*_23_, and *R*_13_ and those synthetized from the sensor model. Furthermore, the estimated parameters have been employed to evaluate the self-compensation method in reducing the position estimation error and the effectiveness of alarm thresholds *T*_1_ and *T*_2_ in detecting wear failure.

### 3.1. Simulation Results

Simulations have been performed in MATLAB 2022b to repeatedly estimate the model parameters *R*_c_ and ξ from a set of resistances *R*_12_, *R*_23_, and *R*_13_ corrupted with numerical noise. The parameter mean values μ and standard deviations σ have been calculated from the estimates across the repetitions taken at different positions.

The resistances *R*_12_, *R*_23_, and *R*_13_ forming the resistance set ***S*_R_** have been synthetized from their respective analytical expressions reported in Equations (3)–(5), fixing the ideal parameters $\bar{p}$ = $\left[\bar{R_{c}},\bar{\xi }\right]$ = [100 Ω; 50,000 Ω/m], the sensor stroke *L* = 100 mm and sweeping the displacement from *x =* 0 to *x* = *L* with a step of 1 mm. The values chosen for $\bar{p}$ are representative of the actual sensor that has been experimentally tested as detailed in Section 3.2. In particular, $\bar{R_{c}}$ = 100 Ω includes non-idealities like the conductive track resistance and the resistance of the connecting terminals. A normally distributed pseudorandom numerical noise with a standard deviation σ_n_ = 1 Ω has been added to corrupt *R*_12_, *R*_23_, and *R*_13_.

For each considered cursor position *x*, the parameters have been estimated from the corresponding resistance set ***S*_R_**. Then, the mean values μ and standard deviations σ with respect to *x* have been calculated from the estimates. The obtained results are reported in Table 2 as the difference between $\bar{p}$ and the estimated parameter mean values μ, along with their standard deviations σ. Figure 4 and Figure 5 report the parameter estimation errors as a function of the position along the sensor stroke.

The simulation evaluates the variability of estimated parameters with respect to the measurement accuracy of resistances *R*_12_, *R*_23_, and *R*_13_. The parameter estimation is adequate for the application scope because the percentage deviations between the ideal parameters $\bar{p}$ and the mean values of the estimated parameters are less than ±0.1%. Furthermore, the parameter standard deviations σ(*R*_c_) and σ(ξ) agree with the corresponding composite uncertainties *u*_c_(*R*_c_) = $\sqrt{3}$/2 Ω and *u*_c_(ξ) = 10 Ω/m, obtained from Equations (8) and (9) by setting *u*(*R*_12_) = *u*(*R*_23_) = *u*(*R*_13_) = σ_n_ = 1 Ω.

### 3.2. Experimental Setup

The DUT is a resistive linear displacement sensor (Gefran PK), with a displacement full-scale span (FSS) *L* = 100 mm, a nominal resistive track resistance of 5 kΩ and an independent linearity error of ±0.05% FSS.

Sensor wearout has been intentionally provoked by sliding the cursor back and forth throughout the whole sensor stroke for 10^6^ cycles. Figure 6 shows the stress-test machine adopted to induce wear, while Figure 7 shows the wearout effect produced on the resistive track.

The assembled experimental setup comprises a linear positioning stage, a digital multimeter, and a personal computer, as shown in Figure 8 and Figure 9.

The sensor parameters are estimated from the resistance sets ***S*_R_** measured in *M* different cursor positions along the sensor stroke. Resistances have been measured by means of a 6.5-digit digital multimeter (Keithley DAQ6510, Cleveland, OH, USA) equipped with a multiplexer module (Keithley 7700), providing a specified resistance measurement accuracy *u*(*R*) = 1 Ω in the 10 kΩ range of interest.

The sensor cursor displacement has been set by a precision linear positioning stage (Physik Instrumente LS-270, Karlsruhe, Germany) with a calibrated position accuracy in the order of ±1 µm. A LabVIEW script has been developed to implement the parameter estimation method and handle the experimental setup. The script is designed to move the linear stage to *M* different positions and trigger the digital multimeter to measure the components of ***S*_R_** at each position.

### 3.3. Experimental Results

Experimental tests have been carried out to estimate the parameters of an undamaged (U) sensor and a faulty (F) sensor, denoted as ***p*_u_** = [*R*_cu_, ξ_u_] and ***p*_f_** = [*R*_cf_, ξ_f_], respectively. For the undamaged and faulty sensors, the respective resistance sets ***S*_Ru_** and ***S*_Rf_** have been repeatedly measured *M* times for the position *x* ranging from 0 to *L* with a 1 mm step resulting in *M* = 100, as schematized in Figure 10. From each set ***S*_Ru_** and ***S*_Rf_**, the parameters ***p*_u_** and ***p*_f_** have been estimated according to Equation (7b).

Table 3 reports the mean values **P_u_** = [μ(*R*_cu_), μ(ξ_u_)] and **P_f_** = [μ(*R*_cf_), μ(ξ_f_)], and standard deviations **σ_u_** = [σ(*R*_cu_), σ(ξ_u_)] and **σ_f_** = [σ(*R*_cf_), σ(ξ_f_)] for ***p*_u_** and ***p*_f_** obtained over the *M* repetitions at varying *x*. From **P_u_** and **P_f_** inserted into Equations (3)–(5), the terms **R_mu_** = [*R*_12mu_, *R*_23mu_, *R*_13mu_] and **R_mf_** = [*R*_12mf_, *R*_23mf_, *R*_13mf_] have been obtained, where the subscript *m* indicates outcomes from the model. Hence the model residuals have been computed for the undamaged and faulty sensor as **r_u_** = [r_12u_, r_23u_, r_13u_] = ***S*_Ru_** − **R_mu_** and **r_f_** = [r_12f_, r_23f_, r_13f_] = ***S*_Rf_** − **R_mf_**, respectively. The residuals **r_u_** and **r_f_** have been evaluated for all the *M* repetitions at which ***S*_Ru_** and ***S*_Rf_** have been measured. The root mean square errors (RMSE) of the residuals are also reported in Table 3. Figure 11 shows the estimated parameters ***p*_u_** and ***p*_f_** while Figure 12a,b report the residuals **r_u_** and **r_f_** as a function of repetition index ranging from 1 to *M* = 100.

The position estimation error *e*_pos_ is evaluated for both the U and F sensors, for all the cursor positions at which ***S*_Ru_** and ***S*_Rf_** are measured. The true, i.e., actual, cursor position *x* is assumed to be given by the linear positioning stage taken as the reference. The position estimations *x’* are derived from Equation (10) by employing μ(ξ_u_) and μ(ξ_f_) for the U and F sensor, respectively. Figure 13 shows the position estimation error *e*_pos_ for both the U and F sensors as a function of *x*. Figure 14 shows *e*_pos_ obtained for the faulty sensor both with and without the application of the self-compensation method described in Section 2.3, where the uncompensated cursor positions are intentionally estimated using μ(ξ_u_) in order to simulate a mismatch between the nominal, i.e., initial, and the current sensor conditions.

Lastly, the fault detection method discriminates the fault and no-fault conditions on the basis of the parameter variations **Δ*p*_u_** = |***p***_u_ − **P**_u_| and **Δ*p*_f_** = |***p***_f_ − **P**_u_|. The fault detection thresholds *T*_1_ and *T*_2_ have been set based on the ***p***_u_ standard deviations **σ_u_**, resulting in *T*_1_ = 3σ(*R*_cu_) = 13.5 Ω and *T*_2_ = 3σ(ξ_u_) = 3.9 Ω/m.

## 4. Discussion

Considering the simulation results described in Section 3.1, the normally distributed pseudorandom numerical noise added to *R*_12_, *R*_23_, and *R*_13_ and the resulting parameter variability are reflected in the parameter composite uncertainties *u*_c_(ξ) and *u*_c_(*R*_c_) expressed in Equations (8) and (9). Assuming the resistance uncertainties equal to the noise standard deviation, i.e., *u*(*R*_12_) = *u*(*R*_23_) = *u*(*R*_13_) = σ_n_ = 1 Ω, the parameter composite uncertainties result *u*_c_(*R*_c_) = $\sqrt{3}$/2 Ω and *u*_c_(ξ) = 10 Ω/m., which agree with the simulation results σ(*R*_c_) = 0.8 Ω and σ(ξ)=11 Ω/m.

The standard deviations of the parameters *R*_c_ and ξ can be analyzed to compare the variability of the parameters obtained in the simulations with those from experimental results. Although the noise standard deviation σ_n_ in simulations is taken equal to the resistance measurement uncertainty *u*(*R*), i.e., *u*(*R*) = σ_n_ = 1 Ω, the experimental standard deviations of *R*_c_, σ(*R*_cu_) = 4.5 Ω and σ(*R*_cf_) = 21.9 Ω, are larger than σ(*R*_c_) = 0.8 Ω for simulations.

This is consistent with the fact that σ(*R*_cu_) and σ(*R*_cf_) cannot be imputed exclusively to *u*(*R*), in fact they can be ascribed to the nonideality of *R*_12_ and *R*_23_ trends as a function of *x* due to the presence of faults, described by the residuals r_12_ and r_23_. Similarly, the residual r_13_ explains the difference between the experimental standard deviations σ(ξ_u_) = 1.3 Ω and σ(ξ_f_) = 9.9 Ω with the simulated one σ(ξ) = 11.5 Ω.

Considering the experimental results described in Section 3.3, Figure 11 shows a visible dependence of *R*_cf_ on the repetition index and, in turn, versus position *x*. Such dependence shows nonuniform variations along the resistive track length *L*, with regions where the contact resistance between the sliding cursor and the resistive track is higher. The microscope analysis confirms that the regions where the contact resistance is higher match with the regions where the resistive track is mostly worn. Therefore, the proposed method theoretically allows detection of local faults through *R*_c_ trend analysis.

The sensor model goodness of fit is evaluated on the basis of the residuals **r_u_** and **r_f_**. As shown in Figure 12a,b, for the undamaged sensor the RMSEs of **r_u_** are ~0.2% of the resistive track nominal resistance, which is adequately low for fault detection and estimation of position *x’*. For the faulty sensor, the RMSEs of **r_f_** increase compared to the undamaged case due to the trends of *R*_12_ and *R*_23_ as a function of *x* caused by nonhomogeneous resistive track wear along its length.

As shown in Figure 13, the position estimation error *e*_pos_ for the undamaged sensor is approximately constant throughout the sensor stroke, with a mean value of 0.2 mm. For the faulty sensor, *e*_pos_ is an order of magnitude higher spanning a range of roughly 2 mm. As reported in Figure 14, the self-compensation method successfully reduces *e*_pos_ to a span of about 2 mm in the faulty case versus the value of 30 mm obtained intentionally using μ(ξ_u_) to represent the case when the parameter estimates are not updated to the current values.

Given the chosen thresholds *T*_1_ = 3σ(*R*_cu_) and *T*_2_ = 3σ(ξ_u_), the results obtained in applying the fault detection method are shown in the confusion matrices of Figure 15.

The fault detection method has been evaluated using accuracy, precision, and recall performance metrics. Accuracy measures the fault detection correctness, precision evaluates the number of actual faults identified, while recall assesses the effectiveness in detecting faults among all instances of faults. The performance metrics obtained for the resistive track and sliding cursor fault detections are reported in Table 4.

The false negative sliding cursor fault detections are due to the variability of *R*_c_ as a function of *x* caused by nonhomogeneous wear of the resistive track. A possible solution could be acting on threshold values. For example, lowering *T*_1_ to equal σ(*R*_cu_) results in the sliding cursor fault detection recall increasing to 96.0%. However, threshold lowering makes the fault detection method more prone to false positives, leading the precision to drop to 78.6%.

Moreover, the uncertainty of the standard deviations of the parameters in the thresholds *T*_1_ and *T*_2_ has an impact on the accuracy and precision metrics for resistive track fault detection. In particular, accuracy and precision vary within [97.5%; 100%] and [95.2%; 100%], respectively.

## 5. Conclusions

This work introduces a suitably general model for resistive displacement sensors and proposes thereof an innovative fault detection method that can be applied with the sensor in operation providing self-diagnostic capabilities. Based on the output of the self-diagnostic step, a self-compensation method with automatic updating is in turn derived to increase the accuracy in sensors subject to progressive wearout. The proposed model and methods have been validated by both numerical simulations and experimental tests on real devices.

The position is estimated by exploiting the sensor model and the initial value of ξ. The self-compensation method reduces the position estimation error by considering the difference between the current and initial values of ξ, which causes a position estimation error. The self-compensation method successfully reduces the position estimation error *e*_pos_ in the faulty case, intentionally obtained using μ(ξ_u_), from a span of 30 mm to a span of 2 mm. The proposed fault detection method evaluates the progressive sensor wearout or sudden failure by comparing the variation in the current parameters *R*_c_ and ξ with respect to their initial values against the alarm thresholds *T*_1_ and *T*_2_, respectively. The fault detection method, for *T*_1_ = 3σ(*R*_cu_) and *T*_2_ = 3σ(ξ_u_), has shown a resistive track and sliding cursor fault detection accuracy of 99.0% and 95.5%, respectively. Lowering *T*_1_ from 3σ(*R*_cu_) to σ(*R*_cu_) has increased the sliding cursor fault detection recall to 96.0%, involving the precision to drop to 78.6% as the threshold lowering makes the fault detection method more prone to false positives.

The future work will involve the implementation of the proposed methods and their testing in a dynamic industrial process.

## 6. Patents

There is a patent application resulting from the work reported in this manuscript [26].

## Figures and Tables

**Figure 1 sensors-24-02594-f001:**
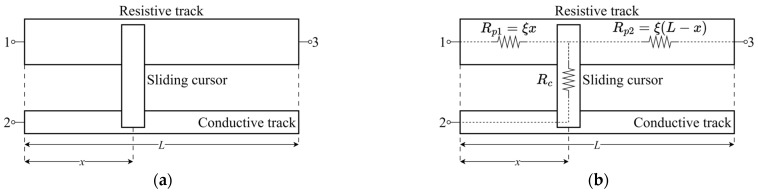
(**a**) Schematization of a generic resistive linear displacement sensor with (**b**) the corresponding resistive circuit modeling.

**Figure 2 sensors-24-02594-f002:**
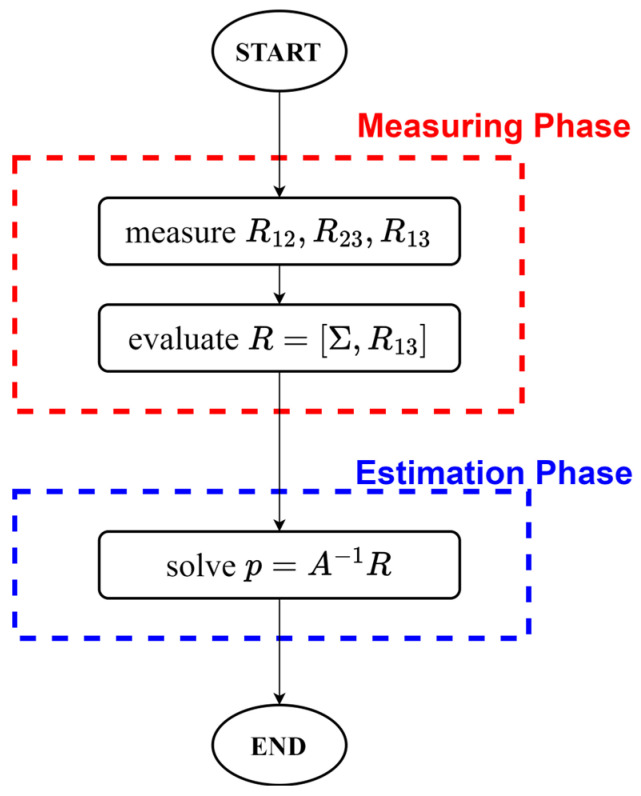
Parameter estimation method flow chart.

**Figure 3 sensors-24-02594-f003:**
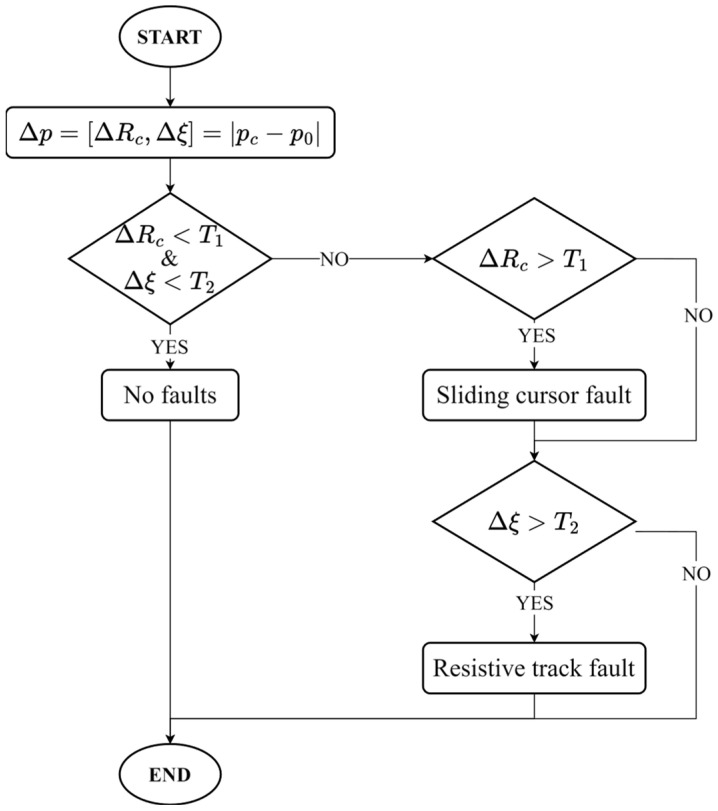
Fault detection method flow chart.

**Figure 4 sensors-24-02594-f004:**
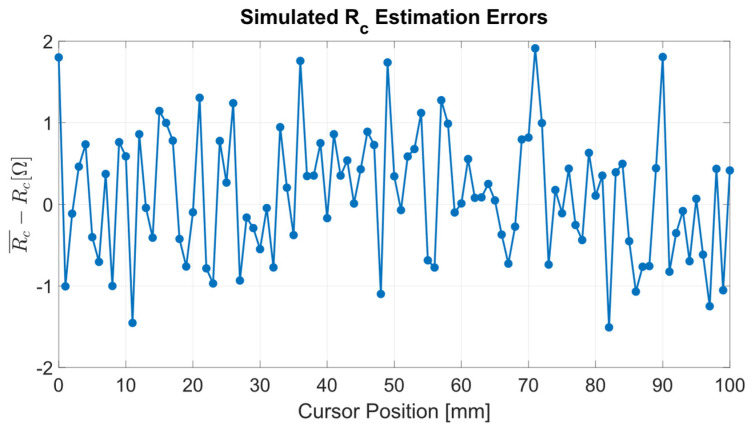
Estimation error of the parameter *R*_c_ versus the position *x* along the sensor stroke.

**Figure 5 sensors-24-02594-f005:**
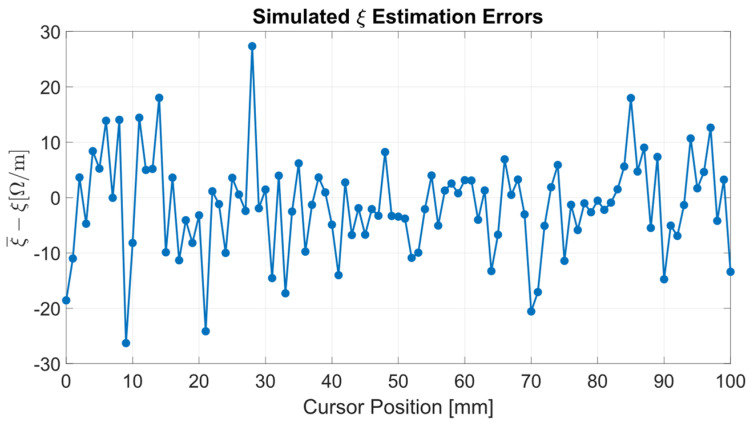
Estimation error of the parameter ξ versus the position *x* along the sensor stroke.

**Figure 6 sensors-24-02594-f006:**
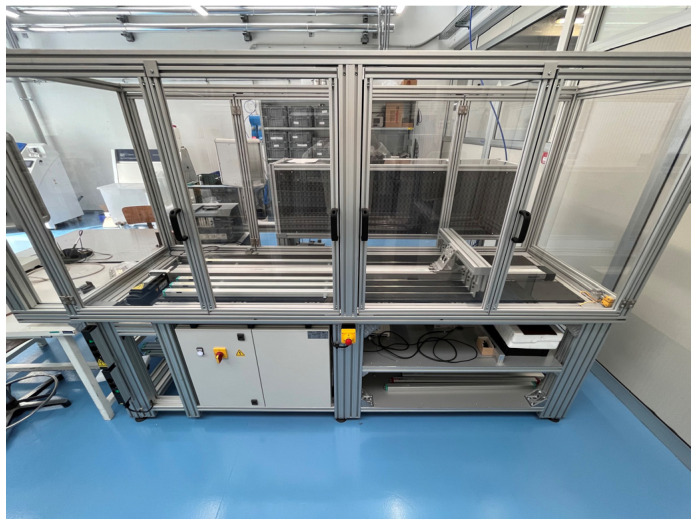
Stress-test machine employed to induce wearout in the resistive track and sliding cursor.

**Figure 7 sensors-24-02594-f007:**
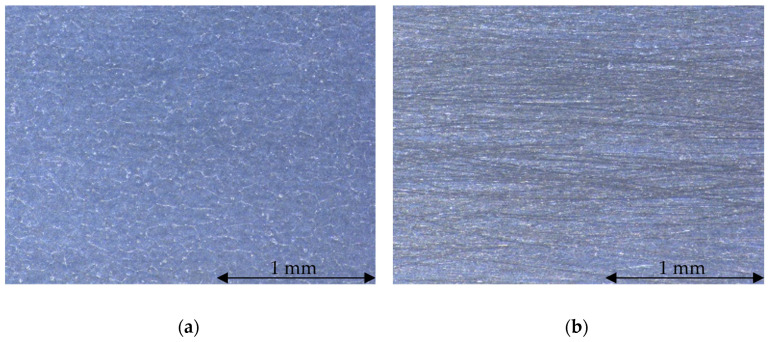
Images obtained with a digital microscopy system (Leica DMS300) of the undamaged (**a**) and wearout (**b**) resistive track.

**Figure 8 sensors-24-02594-f008:**
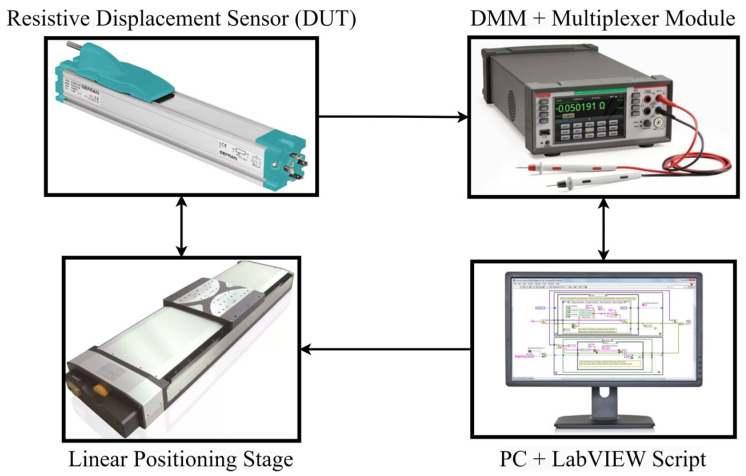
Block diagram of the adopted experimental setup.

**Figure 9 sensors-24-02594-f009:**
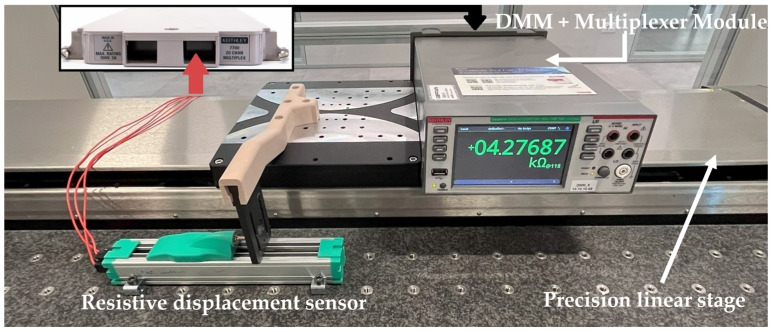
Photo of the assembled experimental setup.

**Figure 10 sensors-24-02594-f010:**
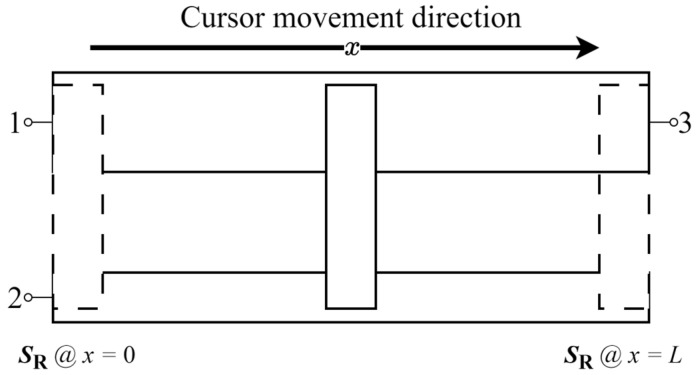
Schematization of ***S*_Ru_** and ***S*_Rf_** measurement processes. The resistance sets ***S*_Ru_** and ***S*_Rf_** are measured from the undamaged and faulty sensor over *M* repetitions taken for *x* varying from 0 to *L* with a 1 mm step.

**Figure 11 sensors-24-02594-f011:**
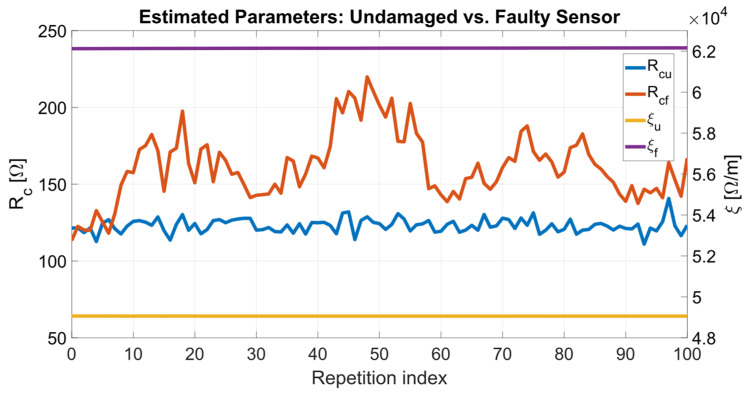
Parameters ***p***_u_ and ***p***_f_ estimated, respectively, from the measured resistance sets ***S*_Ru_** and ***S*_Rf_** as a function of the repetition index ranging from 1 to *M* = 100.

**Figure 12 sensors-24-02594-f012:**
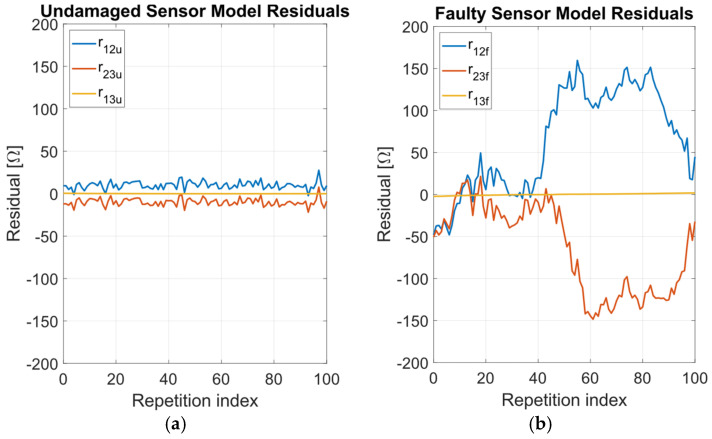
Residuals **r_u_** = ***S*_Ru_** − R**_mu_** (**a**) and **r_f_** = ***S*_Rf_** − **R_mf_** (**b**) resulting from the measured ***S*_Ru_** and ***S*_Rf_** as a function of the repetition index ranging from 1 to *M* = 100.

**Figure 13 sensors-24-02594-f013:**
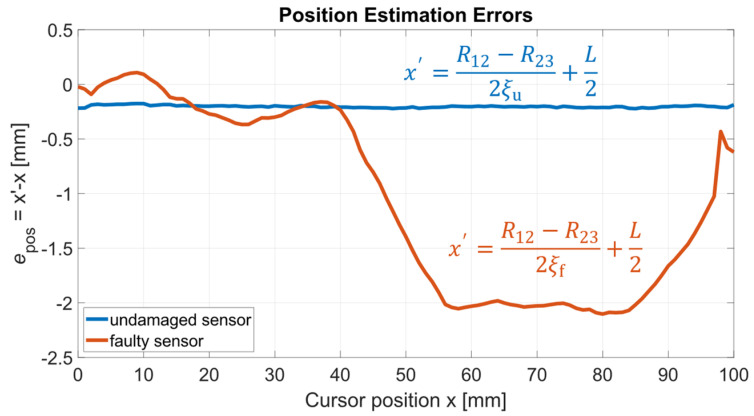
Position estimation error *e*_pos_ for to the undamaged (U) and faulty (F) sensors versus the actual cursor position *x*.

**Figure 14 sensors-24-02594-f014:**
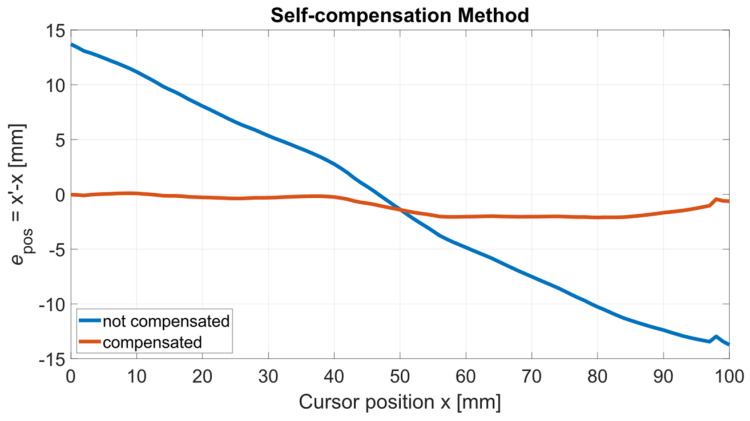
Position estimation error *e*_pos_ for to the faulty (F) sensor, with and without the self-compensation method applied, versus the actual cursor position *x*. The uncompensated cursor positions have been deliberately estimated using μ(ξ_u_) to simulate a discrepancy between the nominal and current sensor conditions.

**Figure 15 sensors-24-02594-f015:**
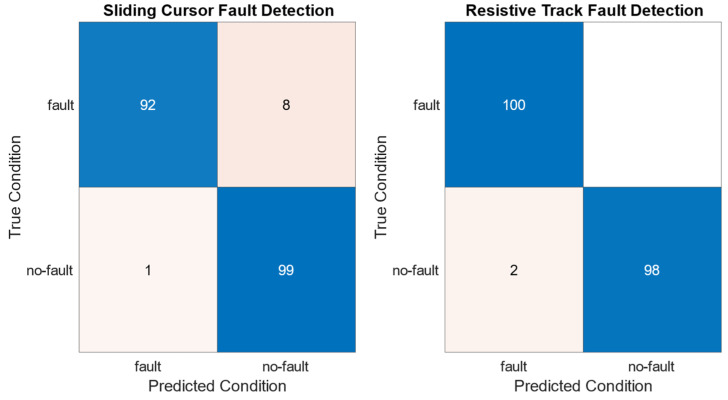
Confusion matrices for the sliding cursor and resistive track fault detections.

**Table 1 sensors-24-02594-t001:** Schematic comparison of the most common methods to detect and identify sensor faults and the approach proposed in this work.

Approach	Description	Properties
Preventive Maintenance	Regular maintenance based on theoretical failure rate to avoid failures before they occur.	It is costly and prone to cause unnecessary repairs.
Analytical redundancy techniques	Detection and identification of sensor failures from models built on a specific process.	It requires deep knowledge of the entire process. The model must be updated as the process changes.
Machine Learning	Detection and identification of sensor faults through models trained on sensor data.	Faults are identified by changes in certain signal characteristic features rather than by underlying sensor-specific causes.
**Method proposed in this work**	Detection and identification of sensor failures from models built on the sensor itself, independently of the processes.	It can be cost effective and the effort for model development can be easily amortized. It offers augmented sensor diagnostic coverage.

**Table 2 sensors-24-02594-t002:** Parameter estimation errors and standard deviations σ obtained by ranging *x* from 0 to *L* with a 1 mm step (*M* = 100 parameter estimations), assuming as a reference the ideal parameters $\bar{p}$ = $\left[\bar{R_{c}},\bar{\xi }\right]$ = [100 Ω; 50,000 Ω/m]. The uncertainties of the parameter mean values μ and standard deviations σ are derived from the parameter composite uncertainties *u*_c_(*R*_c_) and *u*_c_(ξ).

$R_{c}\;\mathrm{Estimation}\;\mathrm{Error}\;\bar{R_{c}}-\mu \left(R_{c}\right)$ [Ω]	$R_{c}\;\mathrm{Standard}\;\mathrm{Deviation}\;\sigma (R_{c})$ [Ω]	$\xi \;\mathrm{Estimation}\;\mathrm{Error}\;\bar{\xi }-\mu \left(\xi \right)[\Omega /m]$	ξ Standard Deviation σ(ξ) [Ω/m]
0.1 ± 0.1	0.8 ± 0.1	−1 ± 1	11.5 ± 0.7

**Table 3 sensors-24-02594-t003:** Parameter mean values μ and standard deviations σ, along with the RMSE of the residuals r_12_, r_23_ and r_13_ obtained in the experimental test. Parameter estimations are performed from the resistance sets ***S*_R_** measured *M* = 100 times for *x* ranging from 0 to *L* with a 1 mm step. Since the resistance measurement uncertainty is *u*(*R*) = 1 Ω, the parameter composite uncertainties are *u*_c_(*R*_c_) = 0.9 Ω and *u*_c_(ξ) = 10 Ω/m. The uncertainties of the parameter mean values μ, standard deviations σ, and RMSE of the residuals r_12_, r_23_ and r_13_ are derived from *u*_c_(*R*_c_), *u*_c_(ξ), and *u*(*R*).

Sensor Condition	*R*_c_ Mean Value $\mu \left(R_{c}\right)$[Ω]	$R_{c}\;\mathrm{Standard}\;\mathrm{Deviation}\;\sigma (R_{c})$ [Ω]	ξ Mean Value μ(ξ) $\left[\Omega /m\right]$	ξ Standard Deviation σ(ξ) $\left[\Omega /m\right]$	$r_{12}$ $\;\mathrm{Root}\;\mathrm{Mean}\;\mathrm{Square}\;\mathrm{Error}\;RMSE\left(r_{12}\right)$ [Ω]	$r_{23}$ $\;\mathrm{Root}\;\mathrm{Mean}\;\mathrm{Square}\;\mathrm{Error}\;RMSE(r_{23})$ [Ω]	$r_{13}$ $\;\mathrm{Root}\;\mathrm{Mean}\;\mathrm{Square}\;\mathrm{Error}\;RMSE(r_{13})$ [Ω]
Undamaged (U)	123.0 ±0.1	4.5 ± 0.1	49,058 ± 1	1.3 ± 0.7	10.9 ± 0.1	10.9 ± 0.1	0.1 ± 0.1
Faulty (F)	161.6 ± 0.1	21.9 ± 0.1	62,142 ± 1	9.9 ± 0.7	88.6 ± 0.1	83.1 ± 0.1	1.1 ± 0.1

**Table 4 sensors-24-02594-t004:** Accuracy, precision and recall performance metrics for the sliding cursor and resistive track fault detections. Accuracy = (TP + TN)/(TP + TN + FP + FN), precision = TP/(TP + FP), and recall = TP/(TP + FN), where TP, TN, FP, and FN are the true positive, true negative, false positive, and false negative fault predictions, respectively.

Type of Fault Detected	Accuracy	Precision	Recall
Sliding Cursor	95.5%	98.9%	92.0
Resistive Track	99.0%	98.0%	100%

## Data Availability

Data presented in this article are available on request from the corresponding author.

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
