# Peer review of "Self-Diagnostic and Self-Compensation Methods for Resistive Displacement Sensors Tailored for In-Field Implementation†"

_sensors, 2024, doi:10.3390/s24082594_

Round 1

Reviewer 1 Report

Comments and Suggestions for Authors

sensors-2924608

Self-Diagnostic and Self-Compensation Methods for Resistive Displacement Sensors Tailored for in-Field Implementation

Indeed, the manuscript is well-written and easy to follow. Some points need to be known.

·         It is better to list a comparison table to compare results with the previous model for resistive displacement sensors.

·         Figure 9, reviewers could not see the connection between DMM and the sensor. Please correct it.

·         Table 1: what are the significance of σ (Rc)=0.8 and σ (ζ)=11.5. Please explain.

·         Table 1 and Table 2: please write the full forms e.g. σ, μ, U, F, RMSE etc.

·         Table 2: What do you menat by RMSE (r13)= 0±1? Please elaborate.

·         Conclusion: “The self-compensation method….”. Please explain, how the proposed method is the self-compensated and self-diagnostic method.

·         It will be good to compare the simulations (MATLAB) and experimental results.

·         It would be good to mention the names of the parts shown in Figure 9.

            ·     The novelty of the work should be clearly highlighted (in the abstract as well as in the conclusion)  

Comments on the Quality of English Language

Minor editing of English language required.

Reviewer 2 Report

Comments and Suggestions for Authors

The manuscript 'Self-Diagnostic and Self-Compensation Methods for Resistive Displacement Sensors Tailored for in-Field Implementation' is focused on actual problem of self-validation of APCS components. The study has interesting results. Firstly, the self-validation topic has little coverage in scientific publications, but at the same time, it is in great demand for industrial applications. Secondly, the authors focused on analytical methods to resolve the self-validation task without ML-methods. Nevertheless, the manuscript will have greater significance if the authors add the results of applying the developed method at an industrial facility.

Reviewer 3 Report

Comments and Suggestions for Authors

This paper presents a comprehensive study on self-diagnostic and self-compensation methods for resistive displacement sensors with a focus on in-field implementation. The manuscript is well-organized, including many valuable materials, data, and analysis. However, before publication, there are several minor comments for consideration:

1. Introduction

It is recommended that the research gap be emphasized and clarified further to underscore the manuscript's contribution to the existing body of knowledge.

2.1. Resistive Displacement Sensors Model

More explanation on the choice of parameters for the sensor model would be helpful. For instance, elaborating on the selection of resistance measurement accuracy and its impact on parameter estimation could provide valuable insights into the methodological decisions made.

2.4 Fault Detection

It would be beneficial to clarify if the features (Rc and ξ) for detection are proposed innovatively by this paper or if they are commonly used ones. If these are commonly used, an explanation of the innovation of this paper in utilizing these features would be informative.

A more detailed explanation of the fault detection thresholds setting and its justification based on experimental data or literature is recommended. The statement that “the threshold can be set empirically” may not suffice; a clearer explanation of how expert knowledge is transformed into an explicit threshold setting would enhance this section.

4 Discussion

Incorporating more comprehensive statistical analysis to validate the experimental results is recommended, such as detection accuracy, false alarm rates, etc. 

Round 2

Reviewer 1 Report

Comments and Suggestions for Authors

sensors-2924608

Self-Diagnostic and Self-Compensation Methods for Resistive Displacement Sensors Tailored for in-Field Implementation

Thank you for allowing me to revise the resubmitted manuscript titled " Self-Diagnostic and Self-Compensation Methods for Resistive Displacement Sensors Tailored for in-Field Implementation." I believe the submitted manuscript and presented work is suitable for publication in Sensors.

Author Response

We appreciate your review and the valuable feedback you have provided. It has helped us enhance the quality of our paper.